# Characterization and Modification of Red Mud and Ferrosilicomanganese Fines and Their Application in the Synthesis of Hybrid Hydrogels

**DOI:** 10.3390/polym14204330

**Published:** 2022-10-14

**Authors:** Arnaldo Ramírez, Leonir Gómez, Alejandro J. Müller, Blanca Rojas de Gáscue

**Affiliations:** 1Laboratorio de Polímeros, Instituto de Investigaciones en Biomedicina y Ciencias Aplicadas, IIBCAUDO “Dra. Susan Tai”, Universidad de Oriente, Cumaná 6101, Sucre, Venezuela; 2Centro de Investigación de Materiales (CIMAT), Universidad Nacional de Guayana, Ciudad Guayana 8001, Bolívar, Venezuela; 3Grupo de Polímeros USB, Departamento de Ciencia de los Materiales, Universidad Simón Bolívar, Apartado 89000, Caracas 1080, Miranda, Venezuela; 4POLYMAT and Department of Advanced Polymers and Materials: Physics, Chemistry and Technology, Faculty of Chemistry, University of the Basque Country UPV/EHU, Paseo Manuel de Lardizabal 3, 20018 Donostia-San Sebastián, Spain; 5IKERBASQUE, Basque Foundation for Science, Plaza Euskadi 5, 48009 Bilbao, Spain

**Keywords:** red mud, modification, hybrid hydrogel

## Abstract

In this work, hybrid hydrogels were synthesized with the inclusion of two types of clay materials that are considered industrial waste: red mud (RM) and ferrosilicomanganese fines (FeSiMn). These solid waste materials were characterized by studying their particle size and chemical composition, which are two key variables for their application in the synthesis of hybrid hydrogels. The morphology imaged by transmission electron microscopy (TEM) and scanning electron microscopy (SEM), showed, in the case of RM, heterogeneous size and shape particles, with 73% of the particles having lengths of less than 5 μm. On the other hand, FeSiMn had particles with a circular morphology of nanometric sizes. Regarding the synthesis of the hybrid hydrogels, it was determined that the incorporation of small percentages (0.1%) of the inorganic phases improved the capacity of the materials to absorb water (swelling indices of 1678% and 1597% for the RM and FeSiMn hydrogels, respectively) compared to the conventional polyacrylamide hydrogel (1119%). An improvement in Vickers microhardness and storage modulus (G′) was also observed: the hybrid with 10% RM presented a G′, 50 times higher than conventional hydrogel. The results show the merit of RM and FeSiMn in improving the properties of hydrogels.

## 1. Introduction

In synthesizing hybrid hydrogel-type composites and nanocomposites, choosing an inorganic phase suitable to obtain the desired properties in the final material is a key step. In this regard, two variables to take into account are: (1) the particle size, which should facilitate good dispersion in the organic matrix and guarantee the homogeneity of the material; (2) the chemical composition, which is responsible for the physical and chemical interactions between both phases that occur during the formation of the hydrogel and determine its final properties. In the following investigation, two inorganic phases were evaluated in the synthesis of hybrid hydrogels, and both considered industrial wastes not studied previously in this type of application: red mud (RM) and ferrosilicomanganese fines (FeSiMn).

RM is a solid residue that is generated from the alumina production process under the BAYER scheme. It is an environmental liability, mainly due to its high alkalinity (pH > 12), which can affect the local ecosystem. The RM used in this research comes from the Venezuelan company CVG BAUXILUM, which produces alumina and generates approximately 1.23 million tons/year of red mud. This residue represents a global environmental liability. It has been reported that the global production of red mud during 2015 reached 4 billion tons/year [1], with an annual increment of 50 million tons [2]. Due to the quantities produced, its handling becomes expensive, and therefore it is deposited in areas called lagoons, which do not represent an integral solution to the problem because sometimes its containment levees have failed, producing spills of this material (the most recent accident occurred in 2010 in Hungary (Ajka), where 1 million m^3^ of mud was spilled, affecting populations and river ecosystems) [3,4,5].

Due to the environmental impact that RM represents and in line with the waste reuse and revaluation policy, in recent years, applications for this material have been searched, focusing on it as an exploitable and usable resource; in this context, the research on red mud has experienced an exponential increase since 2007 [6]. China is the country with the most significant scientific activity on this subject, as it is the largest producer of alumina [1]. The main difficulty in developing industries based on red mud applications is the high cost involved in its neutralization and transport, as it is necessary to lower its pH to a value considered safe between 8.5 and 8.9 [7]. Seawater is commonly used for neutralization, but most bauxite plants are far from the ocean (Venezuela case), making the projects not economically viable. Other neutralization techniques that have been studied include the use of inorganic acids (HCl, HNO_3_, H_2_SO_4_, and H_2_CO_3_). Acid residues from other industries have also been used for this purpose [8,9,10,11], as well as methods based on the use of atmospheric CO_2_ [12] and pyrolysis [13].

Red mud (for its chemical and mineralogical composition) has characteristics that make it attractive in various applications, such as: the catalysis area where it is used for its high content of metal oxides [14,15], as material construction where it can be combined with other materials such as cement [16], in the production of ceramics, and in soil remediation where it is used to adsorb heavy metals [17], in the synthesis of composite materials [18], and as an adsorbent material for aqueous contaminants. In this last application, studies have been carried out in the treatment of water to remove toxic heavy metals, inorganic anions, and metalloid ions, as well as organic dyes, phenolic compounds, and bacteria [6,10,19].

FeSiMn are obtained as a type of waste in the filters of collecting sleeves during the manufacture of Ferro silicon and manganese alloys. These filters are used to prevent the escape into the atmosphere of FeSiMn, emitted by the smelting of minerals in electric furnaces. When FeSiMn powder leaves the production lines, it is stacked and discarded inadequately in the yards at the mercy of wind and rain. This material, due to the volume generated (15 ton/day) is difficult to handle, and its final disposition causes pollutants to the environment. This residue has been little studied in terms of its reuse as a useful resource in other applications.

It is proposed to modify and characterize the two inorganic phases from the mining industry and explore their application in the synthesis of new hybrid hydrogel-type materials to improve the properties of conventional hydrogels and expand their field of applications.

## 2. Materials and Methods

### 2.1. Materials

For all measurements, deionized water was used (17 MΩ, Barnstead-Nanopure deionizer). The RM comes from the CVG BAUXILUM company and the FeSiMn comes from the HEVENSA company, both located in the Guayana region of Venezuela.

### 2.2. Modification of Inorganic Phases

In the literature, the modification of inorganic phases is usually reported as an important aspect that must be evaluated [20]. In this regard, the following two processes were applied to study the influence of the modification of inorganic phases on the properties of hybrid hydrogels.

#### 2.2.1. Neutralization with Synthetic Seawater

To lower the pH of the inorganic phase below 9 units, the inorganic phase was treated with seawater [21]. The prepared seawater was obtained from the Oceanographic Institute of the Universidad de Oriente (Venezuela), and its formulation was achieved by mixing two solutions whose chemical compositions are summarized in Table 1. The treatment consisted of mixing 20 g of the inorganic phase (RM and FeSiMn) with 120 mL of synthetic seawater, with stirring for 6 h. Then, the mixture was allowed to stand for 7 days, monitoring its pH until it reached a value of 8.5. The treated samples were filtered and washed with distilled water. They were subsequently stored in an oven at 60 °C for 1 week.

#### 2.2.2. Acidification with HCl

In total, 10 g of the inorganic phases were weighed, and a 0.1 mol/L HCl solution was added gradually under stirring until pH = 4 was reached. Acidified samples were filtered and washed with distilled water to remove HCl residues. They were subsequently stored in an oven at 60 °C for 1 week.

### 2.3. Determination of Cation Exchange Capacity

The determination of the cation exchange capacity (CEC) was carried out by the method of ammonium acetate [22], in which the ammonium (NH_4_^+^) displaced or replaced all interchangeable cations. Subsequently, the ammonium present at the exchange sites was quantified, and its concentration was expressed in milliequivalent/100 g of sludge (meq/100 g), which represents the CEC.

### 2.4. Granulometry

For granulometry, sieves of the following sizes were used: 150 µm, 75 µm, 38 µm, and 20 µm. The dried red mud was crushed in a mortar, then a 100 g sample was placed in the sieve with the largest mesh size (150 µm) and kept under manual agitation for 20 min. The fraction of the material below the size of the sieve was weighed and placed in the sieve immediately below. The procedure was repeated with the sieves of 75 µm, 38 µm, and 20 µm.

### 2.5. Analysis by Fourier Transform Infrared Spectroscopy (FTIR)

To study the composition and determine the chemical groups present in the structures of the inorganic phases (RM and FeSiMn), they were characterized by Fourier transform infrared spectroscopy (FTIR). For this, the KBr tablet method was used. The initial KBr/RM ratio was 450 mg/15 mg; however, this ratio was increased until spectrum transmittance was improved. The tablets were analyzed in a Perkin Elmer FTIR infrared spectrophotometer, Frontier Optical model (Hopkinton, MA, USA), operating at 20 scans at 2 cm^−1^. This same method was also used to study the interactions between inorganic phases and polymer chains in hybrid hydrogels.

### 2.6. Analysis by Wide Angle X-ray Diffraction

For the identification of the minerals present in the inorganic phases, the Phillips diffractometer model PW 1840 was used (USA), with Cu cathode (Kα, with λ = 1.5405 A); the spectra were taken in the intervals of 5° to 90° (2θ) at a step of 0.02° (2θ).

### 2.7. Analysis by Optical Emission Spectroscopy with Inductively Coupled Plasma (ICP-OES)

For the preparation of the RM and FeSiMn samples, the microwave-assisted digestion technique was used, using the Microwave Digestion System START D equipment from the Milestone (Bergamo, Italy). The chemical composition was determined on an inductively coupled plasma optical emission spectrometer, ICP-OES Optima 5300 DV PerkinElmer brand (Hopkinton, MA, USA). The EPA 3051 method [23] was used. The sample was digested in two stages of 10 min each, with an operating power of 1000 w and a temperature of 175 °C. The elements to be determined by the ICP-OES were Cadmium, Calcium, Cobalt, Copper, Iron, Magnesium, Nickel, Titanium, Molybdenum, Manganese, Selenium, Phosphorus, Vanadium, Lead, Chrome, Zinc, Tin, and Lithium.

### 2.8. Scanning Electron Microscopy Analysis (SEM)

The inorganic phases were characterized by scanning electron microscopy, in an electronic analytical field emission microscope model S-800 FE, brand Hitachi (Tokyo, Japan), operated at 30 kev. The micrographs obtained were studied using the Digimizer software (version 5.8.0, MedCalc Software, Ostend, Belgium), using descriptors of shape and size to characterize the particles.

To study the porosity, the hydrogels were exposed to deionized water, which they absorbed until reaching physicochemical equilibrium. The samples were observed in the Quanta FEG 250 electron microscope (Saragossa, Spain), operated under low vacuum.

### 2.9. Transmission Electron Microscopy Analysis (TEM)

Inorganic phase suspensions in ethanol (0.005 g in 5 mL) were placed on prepared gratings. The samples were observed in the Hitachi model H-600 transmission electron microscope (Tokyo, Japan). Operated with an acceleration voltage of 100 kv. The measurements on the scanned images were made using the Digimizer software.

To study the dispersion of clays in the organic matrix, the xerogels were cut into ultrathin pieces (thickness 80 nm), using the Leica EM UC6 ultratamicrotome (Davie, FL, USA), with a diamond blade. All samples were observed in a transmission electron microscope Hitachi model H-600, operated with an acceleration voltage of 100 kv.

### 2.10. Synthesis of Hybrid Hydrogels

Hybrid polyacrylamide hydrogels (PAAm) were synthesized with the modified and unmodified inorganic phases. In all cases, the following procedure was applied: for the syntheses, the inorganic phases were used on a dry basis. The sample was pulverized with a mortar. The necessary amount of the inorganic phase (0.1%, 1%, or 10% of the total mass of the monomer) was dispersed in a volume of deionized water in test tubes, which was placed under ultrasonic stirring for 15 min. Subsequently, the respective amounts of the monomers and crosslinking agent (N, N′-methylenebisacrylamide (NMBA), manufactured by Sigma-Aldrich (Saint Louis, MO, USA)) were dissolved and placed for 15 min in ultrasound to homogenize the system. Finally, the initiator, ammonium persulfate (PSA) manufactured by Sigma-Aldrich (Saint Louis, MO, USA), was added. Once the mixture was homogenized, it was placed under ultrasonic stirring at 55 °C for 3 h. At the end of the synthesis, the hydrogels were purified with successive washes of deionized water (the water was changed until pH = 7 was obtained) and then dried at ambient conditions. Table 2 shows the nomenclature assigned to each product and the conditions used in the synthesis.

### 2.11. Water Absorption in Hydrogels

The swelling index (S) was determined gravimetrically, for which xerogel tablets were taken and submerged in deionized water. Mass was measured at regular time intervals until reaching physicochemical equilibrium. The swelling index was determined by the following equation.
S = (w_t_ − w_0_)/w_0_ × 100,(1)
where w_t_ is the mass of the hydrogel at a given time and w_0_ corresponds to the mass of the xerogel.

### 2.12. Dynamic Rheology of Hydrogels

For the rheological analysis, the tablet-shaped hydrogels were immersed in deionized water until reaching physicochemical equilibrium (the samples had thickness dimensions between 4 and 5 mm and diameters between 18 and 19 mm). The Rheometric Scientific RDA-II (DE, USA) dynamic torque rheometer was used, with a parallel plate geometry (smooth plates, diameter = 5 cm) fixing a separation between plates of 4 mm. Two types of tests were carried out: the first consisted of a deformation sweep at a constant frequency of 10 Hz, to determine the linear visco-elastic range. The second was a frequency sweep from 0.1 Hz to 100 Hz, at a constant strain of 1% (within the linear viscoelastic regime).

### 2.13. Evaluation of the Microhardness of Hydrogels

Vickers microhardness was evaluated using a Shimadzu brand HMV microhardness tester (Kyoto, Japan). The samples were analyzed in the xerogel state, and the indentation was performed using a load of 980 mN for 60 s.

## 3. Results and Discussion

### 3.1. Determination of Cation Exchange Capacity (CEC)

The CEC is the ability of a material to retain, release, and exchange positive ions with its medium. The CEC determined for RM was 18.1 ± 0.8 meq/100 g. Values of 3.7 meq/100 g have been reported for RM samples from Australia and neutralized with seawater; this value was increased to 15.4 meq/100 g after the application of a chemical treatment to improve its ability to adsorb arsenic [24].

In the case of the RM under study, the results are attributed to the fact that it was not subjected to any neutralization process. The CEC was dependent on the pH: at low pH values, the hydronium ions (H^+^) were strongly attached to the surface of the particles of the material, but at high pH their dissociation was facilitated by generating the exchange sites to trap cations [25].

An important variable to establish the possible applications of the material is to know the type of component it releases, to ascertain its toxic and contaminating potential. In this sense, the cations that the RM exchanged with the medium were determined. The results show that the RM exchanged Fe (0.02 ± 0 meq/100 g) and Ca (10.82 ± 0.03 meq/100 g) mainly. Other metals that were present in it and whose toxic nature have been reported (V, Cd, Pb, Mn, Cu, Co, Cr, Ni, Mo, Sn, and Hg) were also evaluated; however, these were not detected in the analysis. It is noteworthy that Ca and Fe are not considered inorganic water pollutants based on environmental regulations issued by the United States Environmental Protection Agency (EPA) [26] and sanitary standards of drinking water quality in Venezuela [27].

### 3.2. Granulometry

Particle size is a key variable in improving the properties of hydrogels. Very large particles will lead to agglomeration and non-uniformity of the material. For this reason, the determination of the granulometry of the inorganic phases is important in the synthesis of hybrid hydrogels.

The granulometric analysis (see Figure 1) revealed that 50% by mass of the red mud had particles larger than 150 µm and that only 0.15% had particles smaller than 20 µm. Generally, particle sizes for red mud of less than 10 µm are reported for 75% of the sample. On the other hand, the FeSiMn exhibited a granulometry whose highest proportion (86%) was below 250 µm, but with a greater tendency towards smaller sizes, showing a high percentage of particles below 75 µm, which gives an idea about its high mobility in fluids such as air or water.

### 3.3. Analysis by Fourier Transform Infrared Spectroscopy (FTIR)

Figure 2a shows the infrared spectrum obtained for the red mud where the following bands can be highlighted: (1) a band corresponding to the Fe-O bond stretching vibration in the region of 471 cm^−1^; (2) a weak peak around 557 cm^−1^ that has been attributed to the vibration of the Si–O–Al bond [28]; (3) a pronounced band, characteristic of silicate-containing compounds, about 989 cm^−1^, attributed to the Si–O bond, and about 802 cm^−1^ corresponding to the O–Si–O bonds [12]; (4) a small peak around 873 cm^−1^ related to bending movements outside the plane of the CO_3_^2−^ ions of the calcite mineral [6]; (5) a peak around 1387 cm^−1^ that has been attributed to the stretching of the Al–OH bond [28]; (6) the band around 1412 cm^−1^ due to the presence of CaO; (7) a band in the 1473 cm^−1^ region of the anti-symmetric stretch vibration of the C=O bond, associated with carbonates, which form the aragonite mineral [12]; (8) a wide peak attributed to molecular water around 1647 cm^−1^ [29]; (9) a broad band around 3120 cm^−1^ that appears in the characteristic region of the OH bond vibration of bohemite, while the sharp peaks around 3620 cm^−1^ and 3525 cm^−1^ are in the region of the stretching modes of the gibbsite OH bond [8]. These types of bands have also been reported in red mud samples from India [12], and generally in other clays that present some of the oxides found in the RM [30]. The signals corresponding to the bonds between silicon and oxygen are associated with the silanes and silanols groups located on the surface of the RM. These groups represent active sites through which chemical reactions can be promoted, which lead, for example, to obtaining organic/inorganic type compounds.

Figure 2b shows the infrared spectra of the red mud modified with seawater and HCl solution. The main changes observed are the displacements of the signals corresponding to the bonds: Si–O, in the case of HCl modification, observed a split of the signal in two peaks (969 cm^−1^ and 1016 cm^−1^), possibly due to the dissolution of minerals such as sodalite and cancrinite by HCl, which leads to the formation of Si(OH)_4_, generating this new signal at 1016 cm^−1^. The signals corresponding to CaO and Fe–O were also shifted. The characteristic signal of C=O, associated with the presence of aragonite (CaCO_3_), which should be formed by neutralization with seawater was not detected.

The infrared spectrum of FeSiMn is shown in Figure 3a. This inorganic phase has been little studied compared to red mud. No references for the FTIR analysis were found for it. For this reason, the allocation of the bands was completed, taking as reference the general signals that are presented in clays: the wide band observed between 3000 cm^−1^ and 3200 cm^−1^ was indicative of the O–H bond, attributed to the presence of metal hydroxides, as well as to oxidrile groups present on the surface of the clays. The band at 1473 cm^−1^ was attributed to the C=O bond by the presence of carbonates; the signal at 1403 cm^−1^, could correspond to the presence of calcium oxide (CaO). The signal at 1011 cm^−1^ was characteristic of clays containing silicates and was assigned to the Si–O bond that formed the silane group. It has been reported that this bond can generate several observable peaks in the region of 1120 cm^−1^ and 960 cm^−1^ [30], and possibly the signal observed at 1120 cm^−1^ corresponded to the vibration of the Si–O–Si bond. The 493 cm^−1^ and 620 cm^−1^ signals appeared in the region corresponding to Fe–O, Ti–O, Al–O, and Mg–O type bonds [30].

In Figure 3b, the spectra of the modified ferrosilicomanganese fines with seawater and HCl solution are compared. As can be seen, the signal attributed to the Si–O–Si bond disappeared, while the band attributed to the Si–O bond shifted significantly. Other signals, such as the Mg–O (620 cm^−1^) and Fe–O (493 cm^−1^) bonds, shifted slightly.

### 3.4. Analysis by Wide Angle X-ray Diffraction (XRD)

Figure 4 shows the wide angle diffractograms of the red mud before and after being modified; significant differences were observed. In red mud without modifications (Figure 4a), hematite (Fe_2_O_3_) and quartz (SiO_2_) were identified as major minerals, and reflections that have been assigned to calcite (CaCO_3_) (2θ = 29°, d = 3.03 Å) [31], illite ((K,H_3_O)(Al, Mg, Fe)_2_(Si, Al)_4_O_10_[(OH)_2_,(H_2_O)]) (2θ = 8.88°, d = 9.945 Å), goethite (α-FeO(OH)) (2θ = 20.316°, d = 4.368 Å), and cancrinite ((Na,Ca)_8_(Al_6_Si_6_)O_24_(CO_3_,SO_4_)_2_·2H_2_O) (2θ = 24.268°, d = 3.665 Å) also appeared [32]. For the red mud modified with HCl (Figure 4b) the change that stands out is the disappearance of the calcite peak, confirming that the following reaction occured:(2)$$CaCO_{3}+2HCl\to CaCl_{2}+CO_{2}+H_{2}O$$

Similarly, a decrease in the signal that identifies quartz and an increase in the intensities of the hematite and goethite signals were noted. The changes in intensity have been reported related to the amount of crystals present [33], which evidences, together with other results previously discussed, the effectiveness of HCl in modifying red mud.

Figure 5 shows the XRD diffractograms of the ferrosilicomanganese fines and their modifications, and two characteristic signals of the manganese and iron oxide of the jacobsite mineral were observed (Mn^2+^Fe^3+^_2_O_4_) with the following characteristics: 2θ = 29.718°; d = 3.003 Å y 2θ = 32.98°; d = 2.563 Å. This mineral has a cubic crystalline system centered on the faces. In addition, the peak 2θ = 43.4°; d = 2.08 Å could correspond to iron oxide (Fe_3_O_4_). The peak located at 2θ = 61°; d = 1.5291 Å was assigned to iron and copper oxide (CuFe_5_O_8_). Finally, the presence of potassium ammonium sulfate was identified by the peak 2θ = 30.8°; d = 2.893 Å. The diffractograms of the modified fines differ greatly from the unmodified fines, which suggests the efficacy of the treatments applied to this material. In the modification with seawater (Figure 5c), the presence of a characteristic peak of calcite stands out (2θ = 29.4°; d = 3.04 Å), which was formed during neutralization.

### 3.5. Analysis by Optical Emission Spectroscopy with Inductively Coupled Plasma (ICP-OES)

The chemical composition of the red mud determined by the method described in the experimental part is presented in Table 3. Of the elements evaluated, Fe is the one that is found in the highest proportion (between 24% and 27% by mass), followed by the elements Ca, Mg, and Ti. Regarding the minerals that form these elements, it has been reported in samples of red mud [17,34] that the major minerals are hematite (Fe_2_O_3_) and goethite (Fe_(1-x)_Al_x_OOH, X = 0–0.33). Calcium is distributed in minerals such as cancrinite (Na_6_Ca_1,5_Al_6_Si_6_O_24_(CO_3_)_1,6_), calcium oxide (CaO), calcite (CaCO_3_), calcium and aluminum hydrates (x·CaO·yAl_2_O_3_·zH_2_O), and hydrogranate (Ca_3_Al_2_(SiO_4_)_n_(OH)_12-4n_), some of which are included or formed during the alumina production process [14,19]. The oxides of magnesium (MgO) and titanium (TiO_2_) have also been identified, the latter in the form of rutile and anatase.

A chemical analysis revealed traces of elements such as Co, Cu, Ni, Pb, V, and Cr; which have also been detected in other samples of red mud in oxide forms [34]. Minerals such as hematite and geoetite are attributed to a good cation exchange capacity, which generates active sites in the red mud to trap cations from the environment. This enhances their applications as the adsorbent material of cationic contaminants present in soils and aqueous media. In the same way, its application has also been documented as a support for catalysts used in different chemical processes.

On the chemical composition of the ferrosilicomanganese fines, the results show as the major element, manganese (71% by mass), followed in decreasing order by Zn (3.5%) > Ca (2%) > Fe (1.2%). Traces of other elements such as Cu, Ni Cd, V, and Pb were also detected.

### 3.6. Scanning Electron Microscopy (SEM) and Transmission Electron Microscopy (TEM) Analysis

Figure 6a shows the SEM micrographs taken for the red mud. To estimate the average size of the red mud particles, the micrographs were analyzed using the Digimizer software, using size and shape descriptors. The histograms obtained show particle lengths between 0.5 μm and 20 μm. The results also reveal that approximately 73% of the measured particles have lengths less than 5 μm. Liu et al. [32] reported that the typical values for the particle size of RM are less than 10 μm. Other authors in different samples of RM have reported the following sizes: less than 75 μm [28], and less than 5 μm [13].

The same procedure was followed with the ferrosilicomanganese fines; the results are presented in Figure 6b. It was estimated that the fines are formed by particles with a circular symmetric morphology, some with sizes between 40 and 50 nm. However, histograms made from SEM micrographs show that more than 60% of the particles have sizes between 200 nm and 400 nm. This variability of the particle diameters can be attributed to the chemically heterogeneous process that the material underwent, during the fusion of the ferrosilicon, from which they came into atomized form with subsequent crystallization.

Figure 7 shows TEM micrographs of the red mud. It is observed that the red mud has a varied morphology with particles that resemble geometrical figures such as hexagon (Figure 7a) and rectangle (Figure 7b), with average lengths of 2 µm and 1 µm, respectively. Figure 7c shows agglomerates of nanoparticles of different sizes, some with lengths close to 20 nm.

Figure 8 shows TEM micrographs of the ferrosilicomanganese fines, where the uniform circular spherical morphology of nanometric particles can be seen.

### 3.7. Synthesis of Hybrid Hydrogels

The synthesis was conducted by additional polymerization via free radicals, and the formation of the hybrid hydrogels was verified by the change of appearance in the material, which acquired the characteristic color of the inorganic phase used. In contrast, conventional hydrogels are usually completely transparent. It is observed that the final materials have a good homogeneity, which is indicative of a uniform dispersion of the inorganic phase in the polymer matrix (Figure 9). However, a transmission electron microscopy (TEM) analysis of the hybrid hydrogels shows areas with particle agglomeration (see Appendix A).

Obtaining a homogeneous material is related to the interactions that exist between both phases. A good interaction between the monomer and the inorganic phase leads to the stabilization of the pre-synthesis suspension and prevents agglomeration and decantation of the particles. The interactions between both phases are dependent on the pH and the point of zero charge (pHz) of the inorganic phase; with respect to the red mud, a pHz has been reported at approximately 8 [35]. Above this pH value, the red mud particles acquire a negative surface charge as a result of their reaction with the hydroxyl ions present in the water, and the reaction is shown below (S represents the Fe, Si or Al atoms, in RM particle surface):(3)$$S-OH+OH^{-}↔SO^{-}+H_{2}O\;\;\;\;\;\;\;\;(pH>pHz)$$

The pH of the pre-synthesis suspensions (see Table 4) is above the pHz reported for the RM; therefore, it is postulated that the reaction described by Equation (3) occurs. Under these conditions, acrylamide stabilizes the suspension of the inorganic phase through hydrogen bond-type interactions, which would occur between the negatively charged oxygen atoms on the surface of the RM and the hydrogen atoms present in the amide group. These interactions contribute to obtaining a more homogeneous material with better uniformity. A scheme of these interactions is presented in Figure 10.

### 3.8. FTIR Analysis of Hybrid Hydrogels

Figure 11 compares the infrared spectra of the conventional polyacrylamide hydrogel and the hybrid polyacrylamide/RM hydrogel (%RM = 10). The influence of the inorganic phase on the vibration energy of the bonds that form the matrix can be seen. This is manifested with shifts in some signals, which show the types of interactions that are present in hybrid hydrogels and that originate during their formation. Displacements are observed that support the proposal of Figure 10, such as: the deformation of the N–H bond (changed from 1605 cm^−1^ to 1613 cm^−1^); the symmetric stretching of the NH_2_ bonds (changed from 3190 cm^−1^ to 3193 cm^−1^). Other signals remain unchanged such as: C–N bond strain (1412 cm^−1^), asymmetric stretching of the C–C bond (1122 cm^−1^), and the stretching vibration associated with the carbonyl group (C=O) of the amide functions (1654 cm^−1^).

Other researchers who have studied the interactions between polyacrylamide and commercial clays attribute the observed shifts in the characteristic signals of the amide group to the following causes [36,37]:Formation of hydrogen bonds between the surface hydroxyl groups of the clay and the amide group;Rupture of intermolecular hydrogen bonds between polyacrylamide chains, due to their separation;Ion–dipole interactions or coordination bonds, which occur between the exchangeable cations that the clay has and the amide group, for example, between a Ca^2+^ and the free electron pair of the nitrogen atom or the oxygen atom (RM has Ca^2+^ as an exchangeable cation);Formation of hydrogen bonds between the amide group and the interlayer water molecules present in the clay (RM has interlayer water).

Figure 10 also shows the disappearance of some of the characteristic signals of the red mud, such as the one corresponding to the Si–O bond (989 cm^−1^); this type of bond present on the surface of the RM is chemically active, and it has been reported that their participation in chemical reactions lead to the formation of organoclay-type compounds, as well as functionalization reactions [38]. In the same way, the signal that characterizes the silane group O–Si–O (802 cm^−1^) is not observed, whose disappearance could be due to the hydrolysis of this group in the pre-synthesis solution (before starting the polymerization reaction). This reaction leads to the formation of the silanol group (Si–OH).

Figure 12 shows the FTIR spectra corresponding to the hybrid hydrogel (polyacrylamide/FeSiMn) and conventional polyacrylamide hydrogel, and they revealed the interactions that occur between both phases in the hybrid hydrogel, appreciating displacement in the signal associated with the symmetrical stretching of the NH_2_ bonds (from 3189 cm^−1^ to 3182 cm^−1^). In the same way, the band attributed to the Si–O bond (1013 cm^−1^) of the fines present in the hybrid hydrogel is also observed. Other bands remain almost unchanged. The observed displacement suggests the formation of interactions between the ionized surface groups on the fines’ particles (mainly the Si–O^−^ silanol group) and the hydrogen atoms that form the amide group (N–H).

### 3.9. Mechanism of Formation of Hybrid Hydrogels

Based on the information resulting from the FTIR spectra of the materials and in order to explain the observed changes, the following mechanism for the formation of hybrid hydrogels is proposed. For simplicity, it is developed, taking silicon as the main atom on the surface. However, RM is a mixture of minerals that also includes titanium oxide (TiO_2_), alumina (Al_2_O_3_), hematite (Fe_2_O_3_), hydroxides, and carbonates. Likewise, FeSiMn comprises a mixture of minerals and oxides. The mechanism takes place in two stages.

#### 3.9.1. Pre-Synthesis Stage

Initially, there is a dry base red mud with a mixture of silane groups (Si–O–Si) and silanols (Si–OH) on the surface. Upon contact with deionized water in the initial suspension, hydrolysis of most of the silane groups is generated, transforming them into silanol groups (the reaction described in Equation (3) occurs). The pH of the suspension is proportional to the concentration of RM in the system (see Table 4), for the cases studied it is above pHz reported for the RM [35]. Under these conditions the silanol groups ionize releasing a proton. By dissolving the monomers in the system, the RM suspension is stabilized by charge, due to the physical interactions between the ionized silanol groups and the hydrogen atoms of the amide group of the polyacrylamide that have a partially positive charge. These interactions are present in the final material and are evidenced by the displacements suffered by the characteristic signals of the amide group in the FTIR spectra, as previously explained. The reactions involved are presented in Figure 13.

#### 3.9.2. Synthesis Stage

The initiation of the polymerization occurs as in the conventional hydrogel, by hemolytic decomposition of the initiator. Primary radicals are formed that attack the double bond in the acrylamide molecules in suspension, which propagate, occasionally combining with the NNMBA molecules. On the other hand, there are acrylamide molecules physically attached to the surface of the clay, and these molecules can form radical species by their reaction with the initiator or with another growing macro radical (forming active hybrid units, Figure 14). In this way, physically grafted polyacrylamide chains are generated on the RM. When the active hybrid unit reacts during propagation with MBAAm molecules, physical crosslinks are formed between the red mud particles and the organic network. The latter case is schematized in Figure 15.

Figure 16 shows the proposed microstructure for the hybrid hydrogel, with the possible interactions between both phases: (a) physical crosslinking through hydrogen bonds between the negatively ionized silanol group and the hydrogen atoms of the amide; (b) the insertion of polyacrylamide chains to the surface of the inorganic phase; (c) ion–dipole interaction between polyacrylamide and the exchangeable cation (Ca^2+^); (d) hydrogen bonds between polyacrylamide and interlayer water.

### 3.10. Scanning Electron Microscopy (SEM) Analysis of Hydrogels

Figure 17 shows the micrograph obtained by SEM for the conventional polyacrylamide hydrogel. It reveals the porous nature of these materials, and the histogram made from the measurements of the pores shows that approximately 58% of these have sizes less than 6 µm.

In order to analyze the effect of the content of the inorganic phase on the surface morphology and pore size of the hydrogels, the hybrid hydrogels of polyacrylamide/LR and polyacrylamide/FeSiMn were observed by SEM, and the results are shown in Figure 18 and Figure 19, respectively. When comparing the hybrid hydrogels containing red mud (Figure 18a–c) with the conventional hydrogel, it is observed that when the % of RM increases, there seems to be a decrease in the size of the pores, which is reflected with a percentage increase in pore populations with sizes less than 6 µm. This population represents 58% in the conventional hydrogel, and increases to 75% in the PL0 hydrogel, 70% in the PL1 hydrogel, and 74% in the PL10 hydrogel. This decrease in pore size could be due to the ability of RM to form physical crosslinking points in the organic network. The relationship between the pore size of poly(acrylamide-co-itaconic acid) hydrogels and the presence of a physical crosslinking agent such as Ca^2+^ ions has been previously studied and reported [39], showing that the ionic crosslinks generated in the organic matrix can lead to a decrease in the size of the pores. In the case of hybrid polyacrylamide/FeSiMn hydrogels, this increase in the population of the smallest pores is observed more clearly in the hydrogel containing 10% of the inorganic phase (PF10, Figure 19c).

### 3.11. Swelling Index

The swelling index is one of the most important properties of hydrogels, and knowledge of how much solvent it absorbs and how fast it absorbs largely determines the applications of these materials. Figure 20 shows the swelling isotherms for the polyacrylamide/RM (**a**) and polyacrylamide/FeSiMn (**b**) hybrids. In the case of the RM hybrids, it can be seen that for low contents of the inorganic phase (0.1% and 1%), the hybrids outperform the conventional hydrogel; however, when 10% RM is used, the swelling index is similar (within % error) to that of the conventional hydrogel. In the case of the FeSiMn hybrids, the same trend is observed: a decrease in the swelling index proportional to the increase in the concentration of the inorganic phase. The results highlight the dual role of inorganic phases in this type of material, which can provide hydrophilicity through surface groups (for example, the silanol group) that allow greater absorption of the fluid, but in the same way, as the content of the inorganic phase increases, the hydrophilicity is displaced by its ability to form physical crosslinks. This type of physical interaction is evident in the FTIR spectra shown in Figure 10 and Figure 11. These results are in agreement with the structure proposed for hybrid hydrogels and the role played by the inorganic phase in it. This type of trend has been reported in other hybrid hydrogels of polyacrylamide and montmorillonite [40] and polyacrylamide and laponite [41].

When comparing the materials obtained from the modified inorganic phases (Figure 21), it is observed that in all cases, the modifications affect the absorption capacity of the hydrogels, decreasing their hydrophilicity (this was observed regardless of the percentage of inorganic phase used). This may be due to the fact that the surface characteristics of the clays have been altered, causing new interactions between both phases that increase the degree of crosslinking between the polymer chains. Another option is that the modified clays interact with the reagents used in the polymerization (for example, the initiator), affecting the propagation reaction and producing polymer chains of a lower molar mass.

### 3.12. Dynamic Rheology Analysis

Figure 22 shows the results obtained from the rheological analysis of the hydrogels containing RM. As can be seen, the modulus of elasticity (G′) increases with the increasing content of the inorganic phase. The modulus of elasticity is a measure of the stiffness of the material. The results indicate that hydrogels with higher RM content are able to withstand higher stresses without undergoing irreversible deformation. This property is directly related to the structure of hydrogels and the role of the inorganic phase in it. The results can be explained based on the approaches made and the proposed structure: the red mud acts as a multifunctional crosslinker through physical interactions such as hydrogen bonds between both phases. Other authors have also reported that an increase in the content of the inorganic phase leads to an increase in the toughness of hybrid hydrogels. This is due to the formation of agglomerates and the greater number of interactions present between both phases, which facilitate the dissipation of energy, preventing the material from failing [42].

### 3.13. Microhardness Analysis

Material hardness is defined as the resistance to surface deformation. In polymer materials, a relationship has been established between hardness and other properties such as glass transition temperature and storage modulus, since these properties depend on the structure of the material and specifically on intramolecular and intermolecular interactions. Figure 23 compares the Vickers microhardness of the conventional hydrogel and the hybrid hydrogels containing 10% inorganic phase. It is observed that when the inorganic phases are incorporated (modified and unmodified), the Vickers microhardness increases in all cases. This indicates that the hybrids have a more compact structure with a greater number of molecular entanglements or a greater amount of physical crosslinking. Other authors have obtained a proportional relationship between microhardness and clay content in hybrid montmorillonite gels [43]. Likewise, an increase in microhardness has been reported in conventional gels when physical crosslinking was increased [44]. No significant difference is observed between the hybrids synthesized with the different inorganic phases.

## 4. Conclusions

The inclusion of two types of dispersed clay materials that are considered industrial waste, red mud (RM) and ferrosilicomanganese fines (FeSiMn), were considered as fillers for the preparation of hybrid hydrogels.

An extensive characterization of the waste materials was performed by determining their particle size and chemical composition, which are two key variables for their application in the synthesis of hybrid hydrogels. RM contains heterogeneous size and shape particles, with 73% of the particles having lengths of less than 5 μm, while FeSiMn has particles with nanometric circular morphology.

Two chemical modifications were made to the inorganic phases: neutralization with seawater and acidification with HCL and verified by FTIR. CEC measurements showed that RM can exchange cations with the medium, identifying Ca and Fe as the main interchangeable elements.

Homogeneous hybrid hydrogels of polyacrylamide and RM or FeSiMn were successfully prepared. The addition of the unmodified inorganic phases increased the ability of the hydrogels to absorb water, but when the inorganic phases were chemically modified, a decrease in the hydrophilicity of the materials was observed.

The swelling index of the hybrid hydrogels at low concentrations of the inorganic phase (0.1%; 1%) exceeded that of the conventional hydrogel; however, it had a tendency to decrease as the concentration of the inorganic phase increased. These results are attributed to the dual role played by clay materials: they provide hydrophilicity to the material through silanol-type surface groups (Si–OH), but as their concentration increases, hydrophilicity is displaced by their ability to form physical crosslinks in the network, which lead to a progressive decrease in the swelling index.

A dynamic rheology analysis showed an increase in the storage modulus (G′) of the hybrid hydrogels compared to the conventional hydrogel. G′ for the hydrogel with 10% red mud was 50 times higher than G′ for the polyacrylamide hydrogel. This increase is mainly attributed to the physical interactions that are present between the clay and the polymer, which, by forming reversible crosslinking points, act as energy dissipators when the material is subjected to stress, thus preventing it from failing. The improvement in the mechanical properties was also evidenced in the microhardness test, where the incorporation of 10% of the inorganic phase increased the Vickers microhardness between 1.4 and 1.8 times compared to the conventional hydrogel.

The results show the merit of RM and FeSiMn in improving the properties of hydrogels. The hybrids have the potential to be used in the environmental area (removing aqueous contaminants), where they would have better performance than conventional hydrogels (they absorb more fluid), and a longer useful life (they have better mechanical properties). This would reduce the costs involved in its use in real situations.

## Figures and Tables

**Figure 1 polymers-14-04330-f001:**
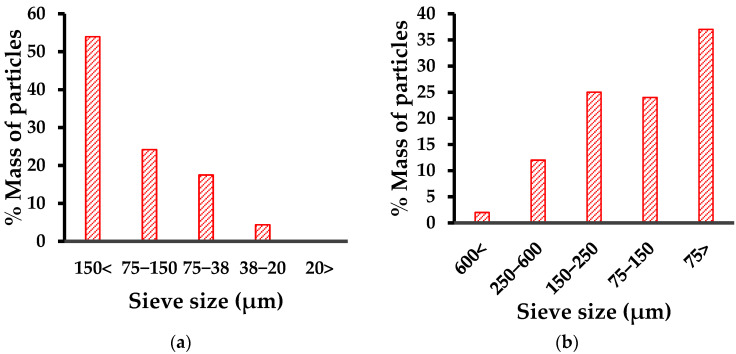
Granulometry for the inorganic phases: (**a**) red mud; (**b**) ferrosilicomanganese fines.

**Figure 2 polymers-14-04330-f002:**
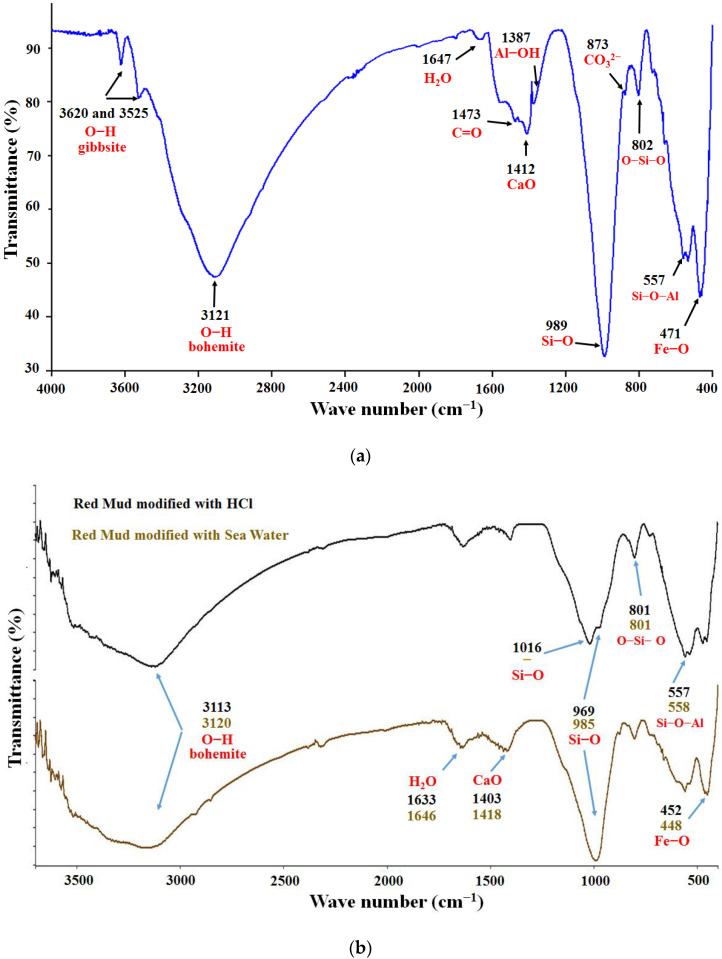
FTIR spectrum of the red mud (**a**) without modification, (**b**) after modifications with seawater and HCl solution.

**Figure 3 polymers-14-04330-f003:**
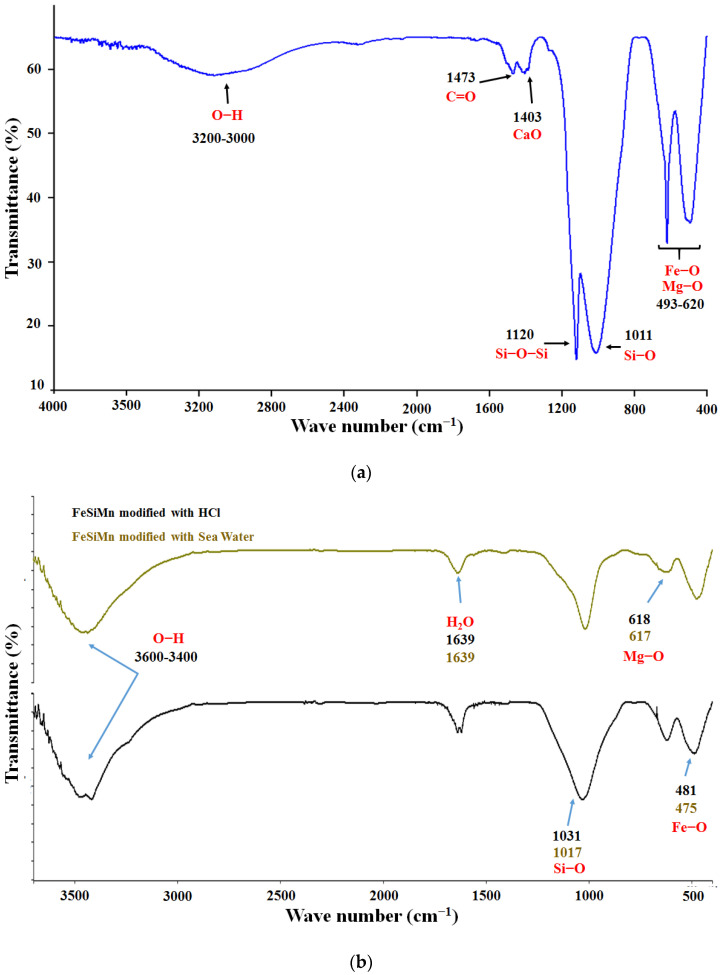
FTIR spectrum of ferrosilicomanganese fines: (**a**) without modification, (**b**) after modifications with seawater and HCl solution.

**Figure 4 polymers-14-04330-f004:**
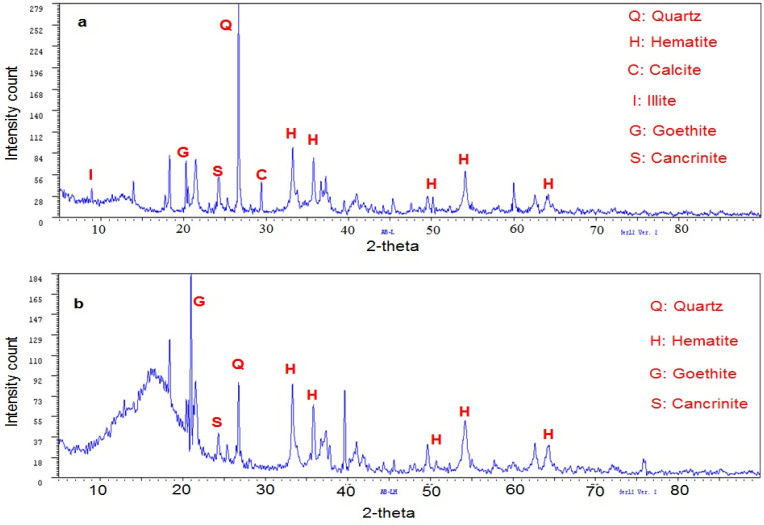
Diffractograms for red mud: (**a**) without modification, (**b**) modified with HCl.

**Figure 5 polymers-14-04330-f005:**
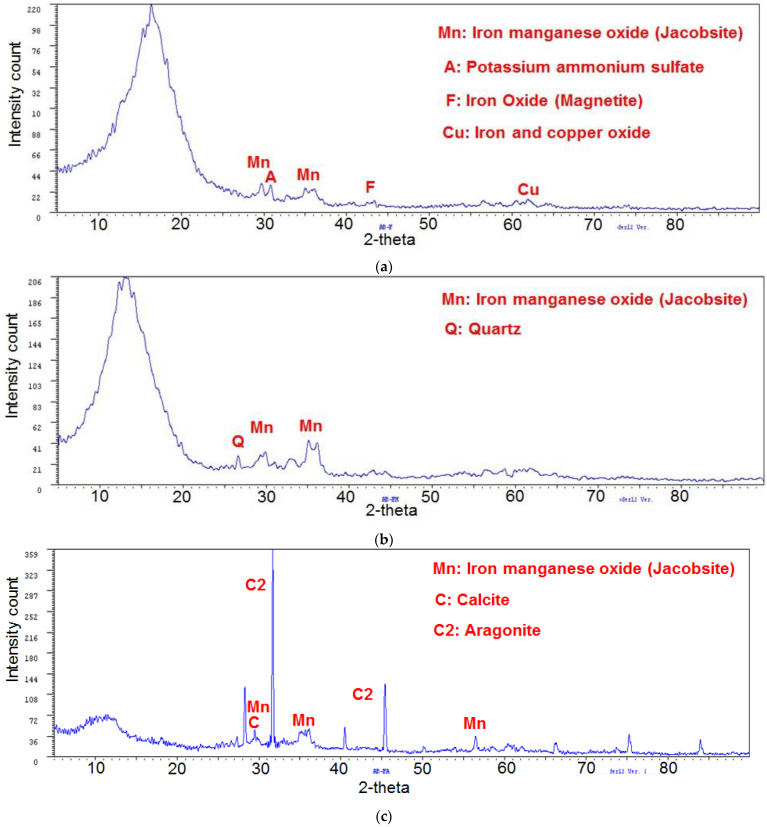
Diffractograms for ferrosilicomanganese fines: (**a**) without modification, (**b**) modified by HCl, (**c**) modified with seawater.

**Figure 6 polymers-14-04330-f006:**
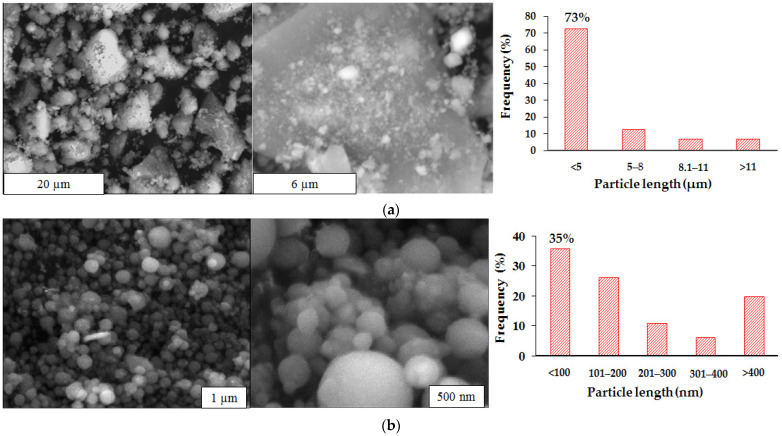
Histograms and micrographs obtained by SEM for the inorganic phases: (**a**) red mud (n = 70); (**b**) ferrosilicomanganese fines (n = 144).

**Figure 7 polymers-14-04330-f007:**
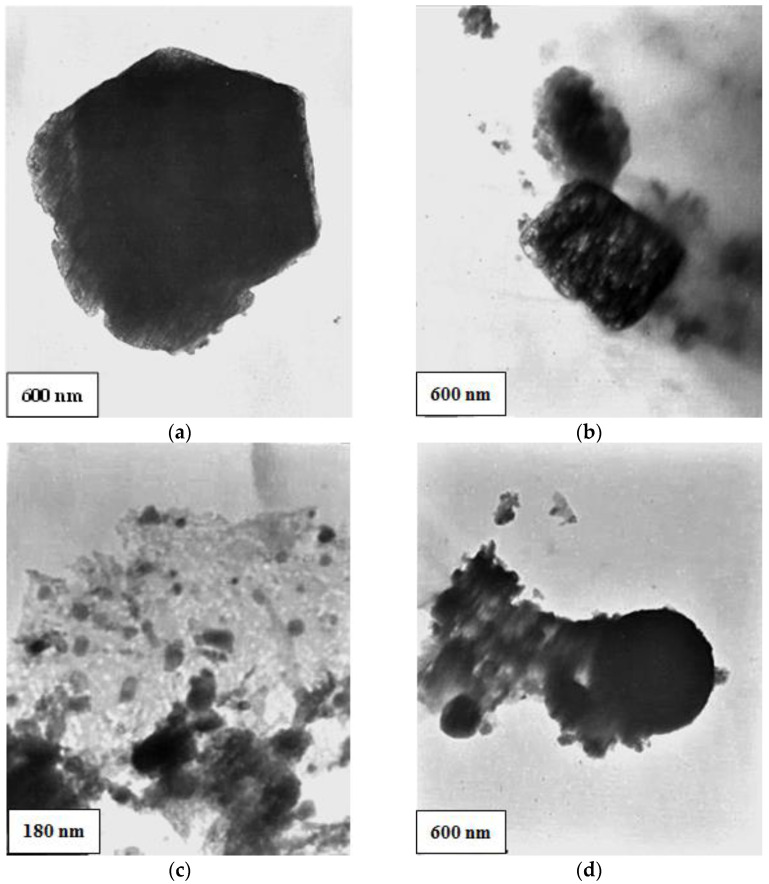
Micrograph by TEM for red mud: (**a**) particle that resembles a hexagon; (**b**) particle that resembles a rectangle; (**c**,**d**) nanoparticle agglomerate.

**Figure 8 polymers-14-04330-f008:**
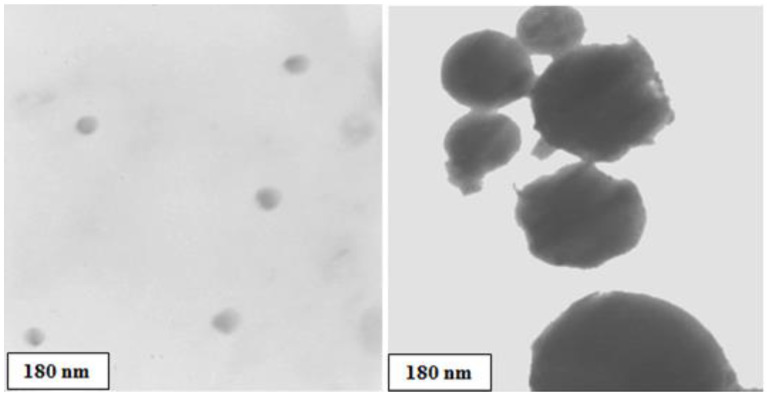
TEM micrographs of ferrosilicomanganese fines. Two types of aggregates of different sizes are shown.

**Figure 9 polymers-14-04330-f009:**
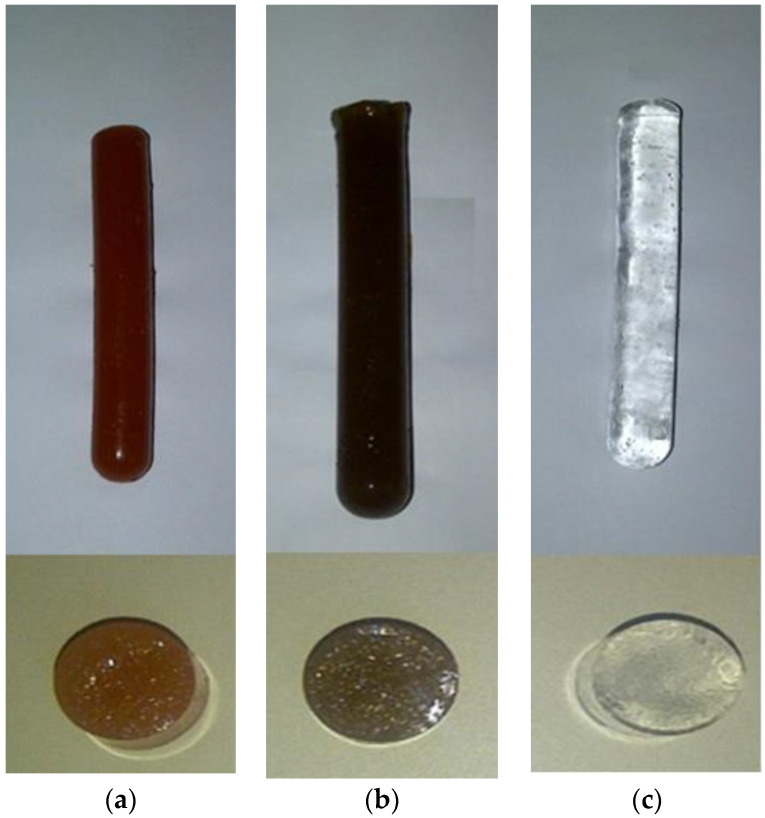
Images of synthesized hydrogels: (**a**) PAAm/RM hybrid HG; (**b**) PAAm/FeSiMn hybrid HG; (**c**) PAAm conventional HG.

**Figure 10 polymers-14-04330-f010:**
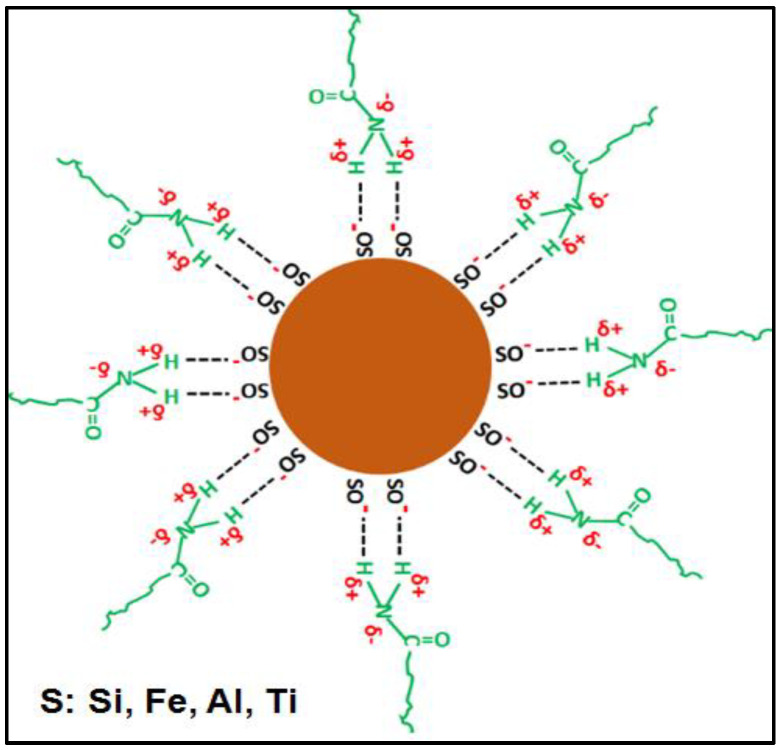
Proposed scheme for the particles of the inorganic phases stabilized by the acrylamide molecules.

**Figure 11 polymers-14-04330-f011:**
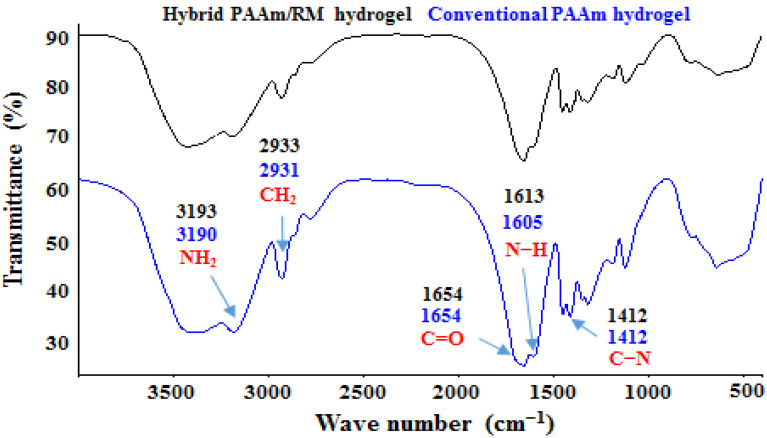
FTIR spectra for polyacrylamide and polyacrylamide/RM hydrogels (%RM = 10).

**Figure 12 polymers-14-04330-f012:**
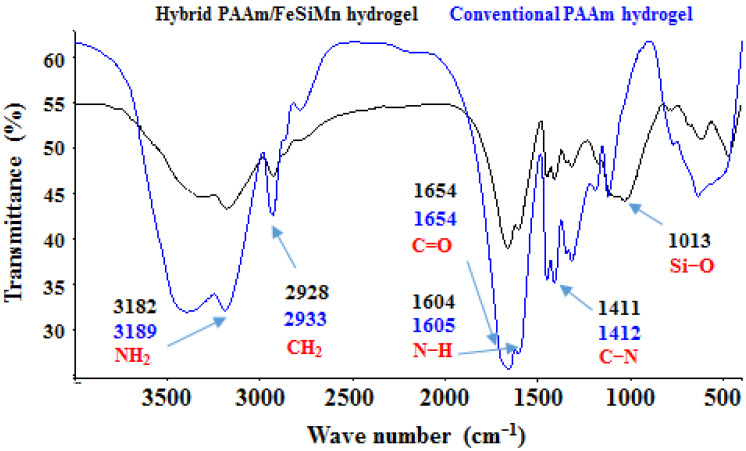
FTIR spectra for polyacrylamide and polyacrylamide/FeSiMn hydrogels (%FeSiMn = 10).

**Figure 13 polymers-14-04330-f013:**
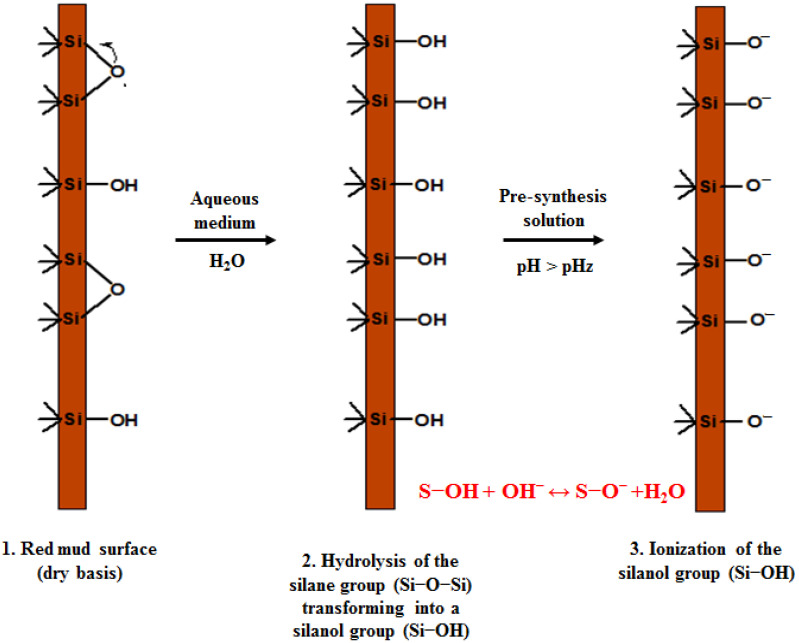
Reactions of the inorganic phases in the pre-synthesis suspension.

**Figure 14 polymers-14-04330-f014:**
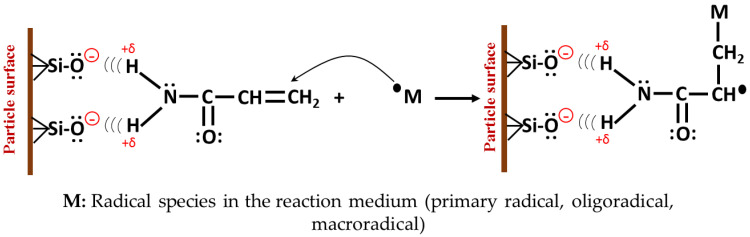
Reaction between an acrylamide molecule physically bound to the surface of the inorganic phase, and a free radical in the reaction medium.

**Figure 15 polymers-14-04330-f015:**
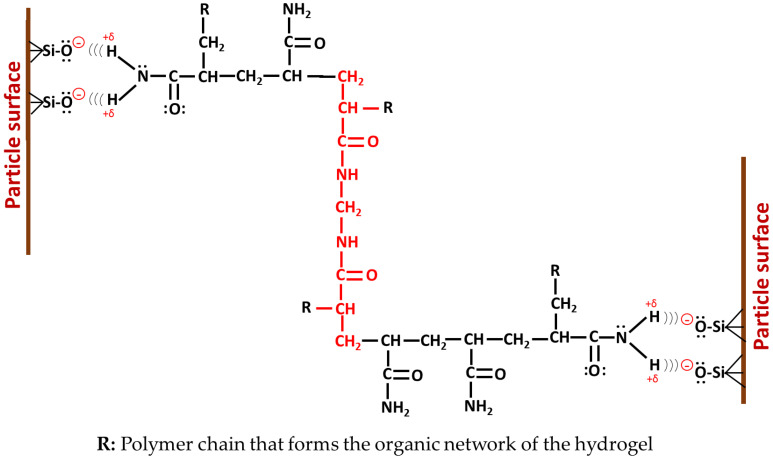
Physical crosslinking between red mud particles and the organic network of PAAm.

**Figure 16 polymers-14-04330-f016:**
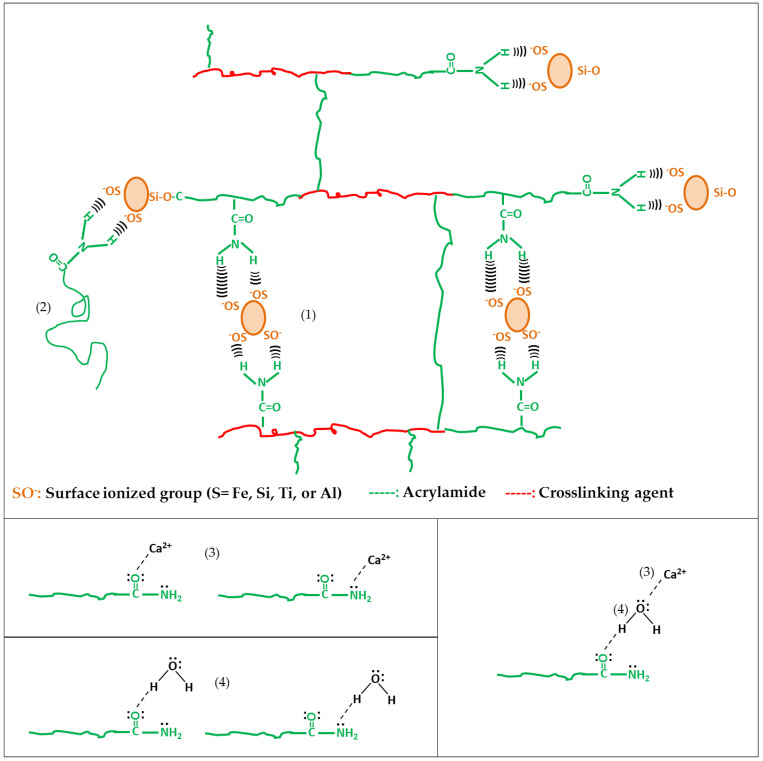
Microstructure for the polyacrylamide/RM hybrid hydrogel and the possible interactions between both phases: (**1**) physical crosslinking by hydrogen bonding, (**2**) physical insertion, (**3**) ion-dipole interaction between polyacrylamide and the exchangeable cation, (**4**) hydrogen bonding between polyacrylamide and interlayer water.

**Figure 17 polymers-14-04330-f017:**
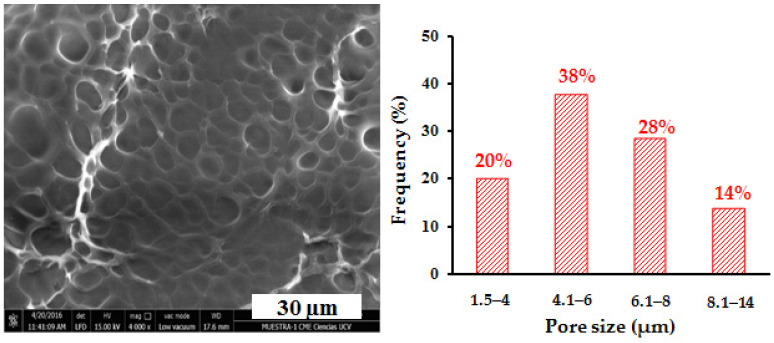
Micrograph obtained by SEM for the conventional polyacrylamide hydrogel (n = 565).

**Figure 18 polymers-14-04330-f018:**
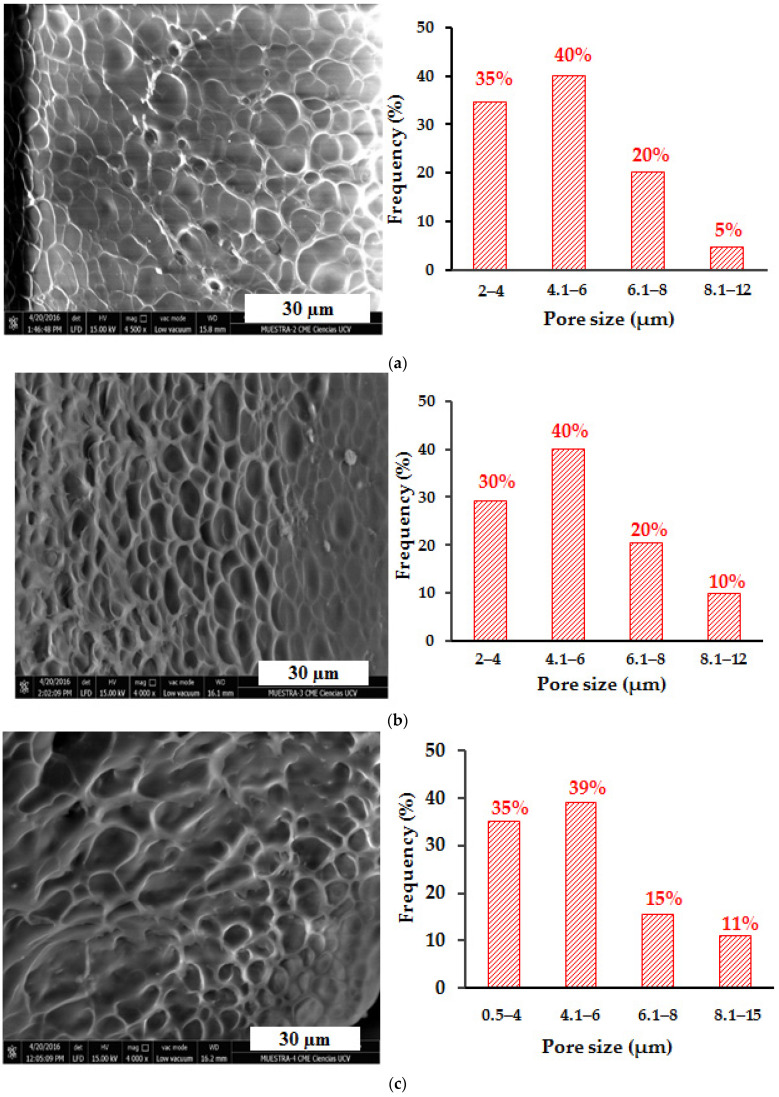
SEM micrographs for polyacrylamide/RM hybrid hydrogels: (**a**) PL0 (0.1%RM, n = 403), (**b**) PL1 (1%RM, n = 361), (**c**) PL10 (10%RM, n = 149).

**Figure 19 polymers-14-04330-f019:**
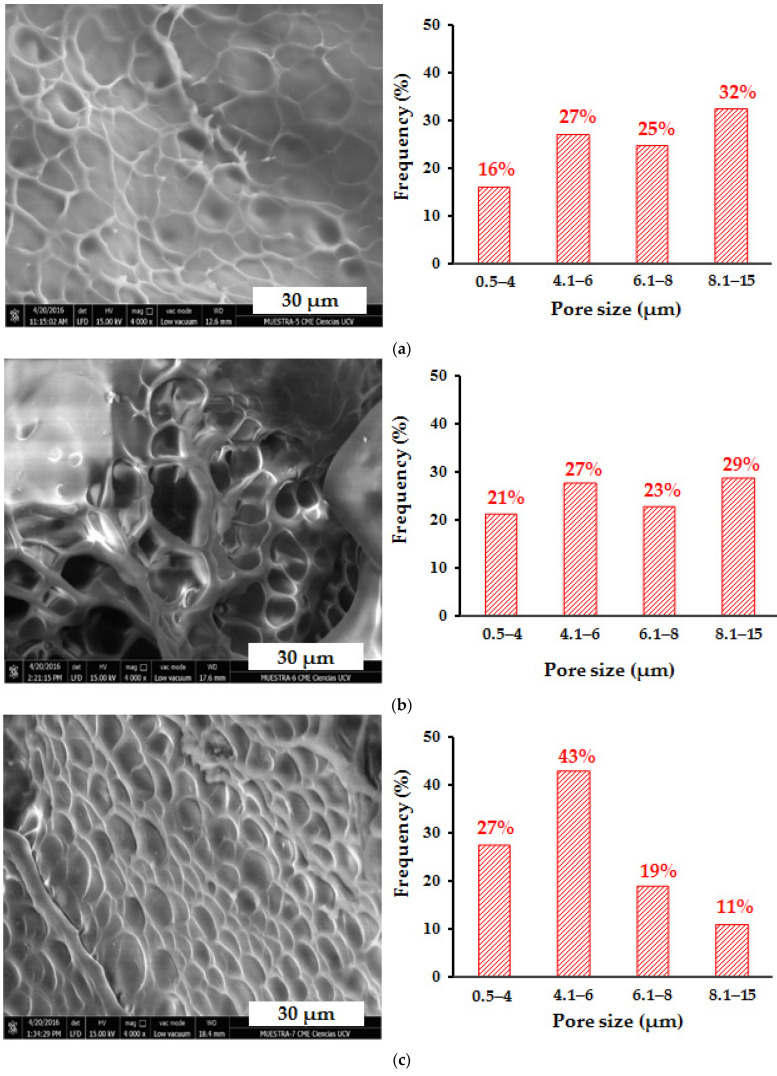
SEM micrographs for polyacrylamide/FeSiMn hybrid hydrogels: (**a**) PF0 (0.1% FeSiMn, n = 519), (**b**) PF1 (1% FeSiMn, n = 203), and (**c**) PF10 (10% FeSiMn, n = 318).

**Figure 20 polymers-14-04330-f020:**
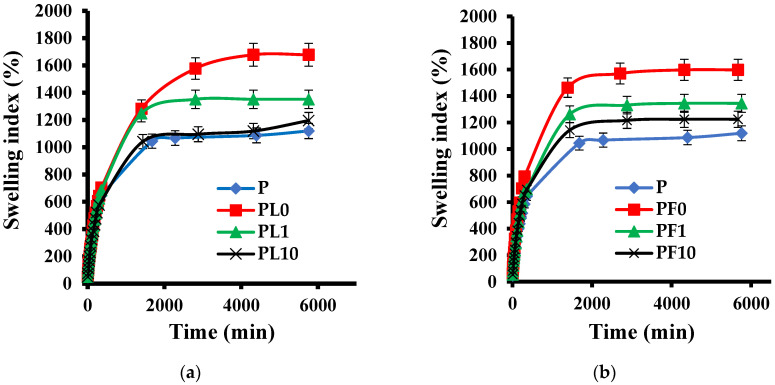
Effect of the concentration of the inorganic phase on the swelling index of hybrid hydrogels: (**a**) polyacrylamide/RM, (**b**) polyacrylamide/FeSiMn.

**Figure 21 polymers-14-04330-f021:**
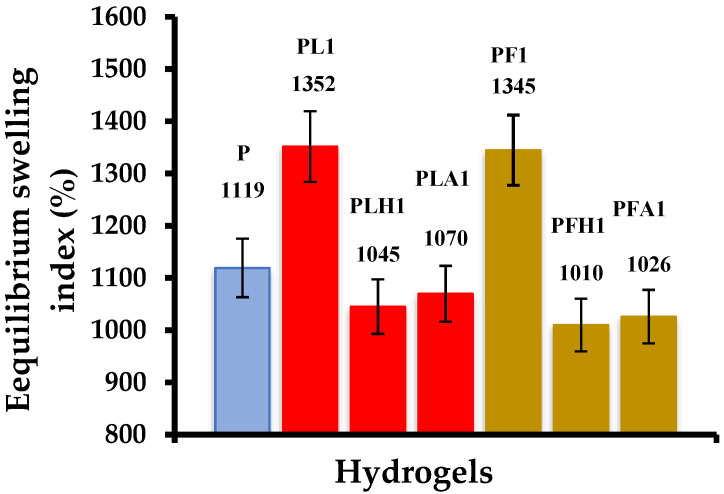
Comparison of equilibrium swelling indices of hybrid hydrogels synthesized from unmodified and modified inorganic phases.

**Figure 22 polymers-14-04330-f022:**
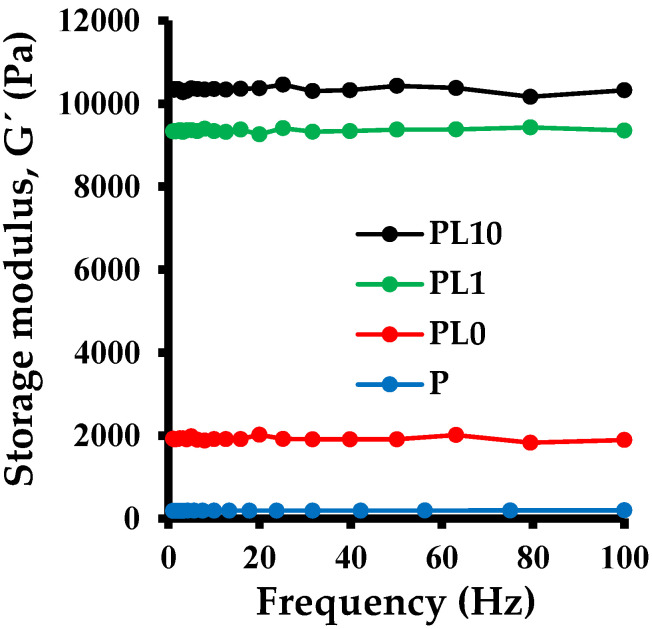
Frequency sweep at constant strain of 2% for hybrid hydrogels with different mass percentage of red mud.

**Figure 23 polymers-14-04330-f023:**
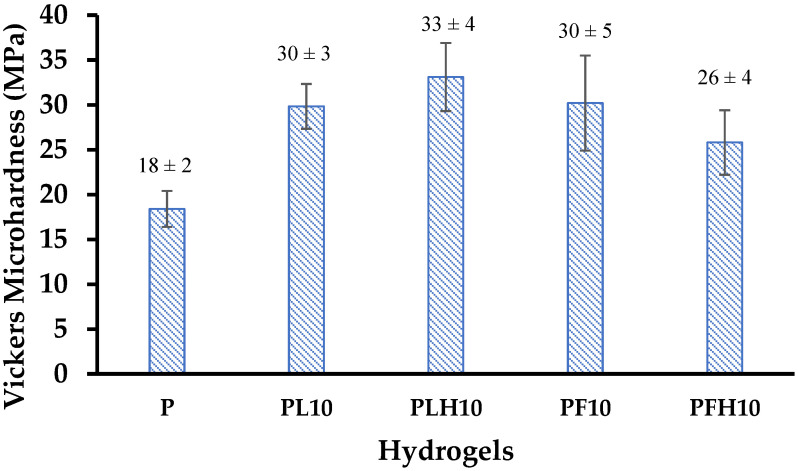
Vickers microhardness of modified and unmodified hybrid hydrogels, with 10% inorganic phase.

**Table 1 polymers-14-04330-t001:** Components of synthetic seawater at a salinity of 35 g/kg.

Solution A		Solution B	
Salts	Mass (g)	Salts	Mass (g)
NaCl	23.9000	Na_2_SO_4_·10H_2_O	9.0600
MgCl_2_·6H_2_O	10.8300	NaHCO_3_	0.2000
CaCl_2_ anhydrous	1.1500	NaF	0.0003
SrCl_2_·6H_2_O	0.0042	H_3_BO_3_	0.0027
KCl	0.6820	Distilled water	100 mL
KBr	0.0990		
Distilled water	856 mL		

**Table 2 polymers-14-04330-t002:** Variables used in the synthesis and nomenclature used in the products.

Mass AAm (g)	Mass NMBA (g)	Mass PSA (g)	Inorganic Phase		Hydrogel Nomenclature
			Type	%	
2	0.02	0.01	–		P
2	0.02	0.01	RM	0.1	PL0
2	0.02	0.01	RM	1	PL1
2	0.02	0.01	RM	10	PL10
2	0.02	0.01	HCl-modified RM	0.1	PLH0
2	0.02	0.01	HCl-modified RM	1	PLH1
2	0.02	0.01	HCl-modified RM	10	PLH10
2	0.02	0.01	Synthetic seawater-modified RM	0.1	PLA0
2	0.02	0.01	Synthetic seawater-modified RM	1	PLA1
2	0.02	0.01	Synthetic seawater-modified RM	10	PLA10
2	0.02	0.01	FeSiMn	0.1	PF0
2	0.02	0.01	FeSiMn	1	PF1
2	0.02	0.01	FeSiMn	10	PF10
2	0.02	0.01	HCl-modified FeSiMn	0.1	PFH0
2	0.02	0.01	HCl-modified FeSiMn	1	PFH1
2	0.02	0.01	HCl-modified FeSiMn	10	PFH10
2	0.02	0.01	Synthetic seawater-modified FeSiMn	0.1	PFA0
2	0.02	0.01	Synthetic seawater-modified FeSiMn	1	PFA1
2	0.02	0.01	Synthetic seawater-modified FeSiMn	10	PFA10

**Table 3 polymers-14-04330-t003:** Chemical composition of the red mud (RM) and the ferrosilicomanganese fine (FeSiMn) determined by ICP-OES.

Element	Wavelength (nm)	RM (mg/kg)	FeSiMn (mg/kg)
Cu	327.393	4.5 ± 0.7	168.6 ± 3.2
Co	228.616	3.5 ± 0.9	ND *
Ni	231.604	2.2 ± 0.4	127.1 ± 1.8
Ti	334.94	1013.9 ± 119	–
Fe	238.204	237,925 ± 15,328	12,015 ± 103
Li	670.784	ND *	–
Cd	214.44	7.0 ± 0.1	71.1 ± 1
Mg	285.213	1140 ± 122.7	–
Ca	213.933	18,453 ± 1568	20,260 ± 1061
Cr	267.716	7.7 ± 1.9	13.1 ± 1.2
Mn	257.61	134 ± 28.6	710,500 ± 35,115
Pb	220.353	37 ± 2.8	488.3 ± 4.4
Mo	202.031	ND *	ND *
P	213.617	15.6 ± 4.6	3.3 ± 0.9
Se	196.026	ND *	71.1±5.3
Zn	213.857	11.3 ± 8.5	35,445 ± 1959
Sn	189.927	ND *	ND *
V	290.88	81.2 ± 4.5	32 ± 0.9

* Not detected.

**Table 4 polymers-14-04330-t004:** pH of the pre-synthesis suspensions used in the synthesis of the hybrid hydrogels.

Suspension	pH as a Function of the Percentage of Inorganic Phase			
	0%	0.1%	1%	10%
Aam + RM	5.84	8.25	9.87	10.47
Aam + FeSiMn	5.84	7.93	9.21	10.20

## Data Availability

The experimental data are reported within the manuscript. Digital version of raw data and figures are available upon request from the authors.

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
