# Peer review of "Characterization and Modification of Red Mud and Ferrosilicomanganese Fines and Their Application in the Synthesis of Hybrid Hydrogels"

_polymers, 2022, doi:10.3390/polym14204330_

Round 1

Reviewer 1 Report

This work investigated the preparation of hybrid hydrogels with the inclusion of two types of clay materials: red mud (RM) and ferrosilicomanganese fines (FeSiMn). The experiment is well designed and the result is well analyzed, I think this work is suitable to be published in Polymers after some revisions as follows:

1. Line 168, “The sample was sprayed to obtain uniform particle sizes.” How to determine that the particle sizes of the sample are uniform, Figure 1 dose not show the uniform particle sizes.

2. Line 374, Figure 11 indicated a uniform dispersion of the inorganic phase in the polymer matrix, more evidence should be given to justify the uniform dispersion of the inorganic phase.

3. The formation mechanism of the hybrid hydrogel should be summarized in the manuscript.

4. There are grammar and format mistakes in the manuscript, please carefully check the whole manuscript and make corresponding revision.

5. The format of citation should be carefully checked and corrected.

Reviewer 2 Report

The purpose of this research is clear and of high interest. However, the discussion is concluded abruptly without touching on the primary objective: hybrid hydrogel in detail. 

Experimental section lacks detail. For example, how many ml of seawater was added /g of the material?

Result and discussion: for ease of comparison, figs 2&3; 4&5 should be combined. 

Fig 6: Why is there no data for the seawater-treated RM?

Fig 7C: please elaborate on how calcite is formed during the seawater treatment.

Hydrogels: what is the density of these hydrogels before water absorption? For enhancing 20% water absorption, how much weight is added?

For establishing the potential application of these muds or silicates, there should be clear information on the merit, and demerit.

Reviewer 3 Report

The manuscript studied the potential of red mud and ferrosilicon as core materials for hydrogels in the application of water treatment. Though the topic itself is interesting, we find that the manuscript cannot be accepted in the current stage for the following reasons:

1.     The main contents of the manuscript are the characterization of the red mud, ferrosilicon, and their treated product. The treated methods are also well-known methods as mentioned in the introduction.

2.     There are almost no studies about hydrogel properties besides the equilibrium index. For submission, further studies of the hydrogel are needed. For example, SEM of hydrogel…

3.     Though the authors successfully fabricated the hydrogels, it is questionable about their application. The authors mentioned the potential for ionic exchange. It is highly recommended for the application of the hydrogel be carefully investigated.

For these reasons, we believe that manuscript is not suitable for the journal.

Round 2

Reviewer 1 Report

The manuscript has been well revised and the paper can be published in Polymers. 

Reviewer 3 Report

After the major revision, the manuscript was significantly improved. With the information rerated to the gel been fully studied, the manuscript’s content is suitable to be accepted by the journal Polymers (MDPI).